# Enzyme Activities in Reduction of Heavy Metal Pollution from Alice Landfill Site in Eastern Cape, South Africa

**DOI:** 10.3390/ijerph191912054

**Published:** 2022-09-23

**Authors:** Nontobeko Gloria Maphuhla, Francis Bayo Lewu, Opeoluwa Oyehan Oyedeji

**Affiliations:** 1Department of Chemistry, Faculty of Science and Agriculture, University of Fort Hare, Private Bag X1314, Alice 5700, South Africa; 2Department of Agriculture, Faculty of Applied Sciences, Wellington Campus, Cape Peninsula University of Technology, Wellington 7655, South Africa

**Keywords:** soil pollution, heavy metals, enzyme activity, enzyme inhibition

## Abstract

Heavy metals are unbreakable, and most of them are poisonous to animals and people. Metals are particularly concerning among environmental contaminants since they are less apparent, have extensive effects on ecosystems, are poisonous, and bioaccumulate in ecosystems, biological tissues, and organs. Therefore, there is a need to use biological agents and phytoremediation processes such as enzymes because they have a high potential for effectively transforming and detoxifying polluting substances. They can convert pollutants at a detectable rate and are potentially suitable for restoring polluted environments. We investigated heavy metal concentrations in different soil samples collected in four sections in Alice and determined the enzyme activity levels present in the soil. The Pearson correlation analysis was conducted to check whether there was any relationship between heavy metal concentrations and enzyme activities in the soil. Samples were randomly collected in three weeks, and the microwave digestion method was used for sample treatment and preparation. Quantitation was achieved by inductively coupled plasma mass spectrometry (ICP-MS). The enzyme assay through incubation method was implemented for discovering the four selected enzymes (urease, invertase, catalase, and phosphatase), and their activity levels were examined colorimetrically by colorimetry spectrophotometer. The ICP-MS results revealed 16 predominating elements, namely: Al, Ba, Ca, Co, Cr, Cu, Fe, K, Mg, Mn, Na, Ni, Sr, and Zn, and the presence of a non-mental, which is phosphorus (P), and a metalloid in the form of silicon (Si) in all soil samples. Significant differences in metal concentrations were observed among the collection sites. The Al, Fe, K, Mg, and Ca concentrations were above WHO’s permissible limits. While Ba, Mn, Na, and P were in moderate concentration, Cu, Cr, Co, Zn, Sr, and Ni were in small amounts recorded mostly below the permissible values from WHO. Four soil enzyme activities were determined successfully (urease, invertase, phosphatase, and catalase). A negative non-significant correlation existed between urease, invertase, phosphatase enzyme activity, and the concentration levels of all selected metals (Al, Ba, Ca, Co, Cu, Fe, K, Mg, Mn, Na, Ni, Cr, Sr, and Zn. In contrast, the content of catalase activity was associated non-significantly but positively with the range of selected heavy metals. This study suggests proper monitoring of residences’ areas, which can provide detailed information on the impact of high heavy metal content on people’s health. They are easily dispersed and can accumulate in large quantities in the soil. The necessary implementation of waste management programs will help the municipality adopt a strategy that will promote recycling programs and protect the residence health from this threat.

## 1. Introduction

Heavy metals are chemical element materials with relatively high densities that exist naturally in numerous amounts in the environment. Most heavy metals are poisonous even in low quantities, and their accumulation in bodily tissues over time may be harmful to human health [1]. Some metals are carcinogenic, genotoxic, or cause genetic mutations in humans and animals depending on the amount and duration of exposure; these include As, Cd, Cr, Cu, Hg, Mn, Ni, Pb, and Zn [2].

Soil health refers to the healthy balance of organisms and their surrounding environment inside the soil ecosystem. Every disruption to the soil caused by the negative impacts of pollutants on soil biochemical activity affects soil health and functions [3]. Soil enzymes are derived mainly from microbes, with some originating from plant or animal wastes. Enzymes accumulate in the soil as free enzymes or enzymes stabilized on clay surfaces and soil organic materials. Most enzymes are often employed to assess the impact of pollutants, such as dehydrogenase (DH), phosphatase (PHO), and urease (UR) [4]. Soil phosphatase is required for organic phosphorus mineralization in soil [5]

According to the evaluation studies conducted in this area, Alice is one of the smallest towns facing health and environmental concerns due to the build-up of overflowing bins and illegal disposal sites, and the management and disposal of hazardous materials [6]. The increasing growth in waste output and inappropriate trash disposal has led to the historical backlog of insufficient waste services, resulting in uncomfortable living circumstances, an unhealthy environment, and heavy metal build-up in nature. Heavy metal pollution is a severe global environmental concern since it contributes to environmental disturbance due to its plentiful sources, non-biodegradable qualities, and accumulative behaviour. Due to their persistence in nature and potential to bioaccumulate, toxic metals can induce enzyme inactivation, resulting in changes in soil properties, productivity limitations, and ecosystem function [7]. 

Even though contamination with heavy metals is a global problem, pollution levels depend on location [2]. The utilization of enzyme activity in the soil to reduce pollution has been applied in other areas worldwide. Still, there is very little or no data for this initiative across the Eastern Cape province and South African regions at large.

Soil enzymes are well-known for accurately reflecting the degree of deterioration of soil quality caused by soil pollution and diagnosing the functional recovery process of polluted soil. So far, contaminated site rehabilitation has mainly focused on pollutant removal, which presents expenses and secondary ecological disturbance in the repair process. Sustainable soil remediation aims to save costs, restore soil health, minimize environmental disruption, and maintain its effects. The study aims to determine enzyme activity in the soil and determine if the available activity of soil enzymes can be utilized to monitor the soil pollution and remediation process of contaminated soil. 

## 2. Materials and Methods

### 2.1. Description of the Study Area

The study was carried out in Alice Township located in Victoria East under the Nkonkobe Municipality situated along the southern slopes of the Winterberg Mountains range and escarpment in the Province of the Eastern Cape, with geographical coordinates of 32°47′0″ S, and 26°50′0″ E.

The sample sites are the Alice landfill site and the East campus inside the University of Fort Hare (as shown on Figure 1). The East campus was used as a control site and is approximately 4 km from the dumping site. The control site (Site 2) found a way down to the bottom of Somgxada hills, alongside the University fencing. The soil is covered by abundant natural vegetation, while Site 1, which is the landfill site, is divided into three portions, namely A, B, and C, Portion A is found on the east side of the dumpsite, where the ground is covered by plenty of rusted and burned tins, broken bottles, or glasses, and rusty wires from car tires. 

In contrast, portion B is situated close to where the trucks and motor vehicles deliver garbage on the west side. Lastly, portion C is sited outside the dumpsite fencing, and many different natural plants cover the surface.

#### 2.1.1. Collection and Preparation of Soil Sample

The soil samples were randomly collected at a 0–25 cm depth twice a week for 3 weeks. The sampling was carried out in sites 1 and 2 using the clean soil auger. The dry soil samples collected were placed in clean, labelled polyethylene bags and then transported to the laboratory for further analysis [8]. The soil samples were grinded using a mortar and pestle to reduce the particle size and then sieved through a 2 mm mesh to obtain ac-acceptable and homogeneous samples. Each sample was divided into two, and the first part was stored at room temperature until the physicochemical parameter analysis was performed. Simultaneously, the other part was stored in the refrigerator at 4 °C until enzyme analyses were performed [9].

#### 2.1.2. Enzyme Assay

The soil samples were incubated in 250 mL flasks at 25 °C and 10% *w/w* water content for 5 weeks. Afterward, aliquots of soils (1 g, 2 g and 5 g) will be incubated in 50 mL flasks for different times (20 min, 1 h, 3 h, 12 h or 15 h), depending on the target soil enzyme assayed. The enzyme activities will be assayed according to the principle of incubating on the table below (as shown on Table 1) [10].

#### 2.1.3. The Inductively Coupled Plasma Mass Spectrometry (ICP-MS) Analysis

The dried homogenized powdered soil samples were analysed using a microwave digestion system. Then, 1.0 g of soil sample was weighed, pre-treated, and digested with 5 mL Nitric acid (HNO_3_) and 5 mL Perchloric acid (HClO_4_). The suspension was allowed to digest, and sample was evaporated on a hot plate to initial dryness. The sample was allowed to cool in a desiccator before ICP-MS analysis [11].

#### 2.1.4. Statistical Analysis

The data were analysed using the IBM Statistical Package for Social Science (SPSS) 26, version 26.0 (IBM, Armonk, NY, USA). Tukey Post hoc tests at *p* ≤ 0.05 determined the multiple comparisons of means from one-way analysis of variance (ANOVA) and the significant difference among the selected enzyme activity means. IBM Pearson’s correlation analysed the relationships between soil enzyme activity and physicochemical parameters. This study was approved for ethical clearance by AREC University of Fort Hare, with certificate number: OYE021SMAP01/19/E.

## 3. Results

### 3.1. ICP-MS Elemental Analysis

The ICP-MS analysis (as presented in Table 2, Figure 2 and Figure 3) shows 16 elements discovered in this study. The results reveal 14 heavy metals, namely: Al, Ba, Ca, Co, Cr, Cu, Fe, K, Mg, Mn, Na, Ni, Sr, and Zn, and the presence of a non-mental, which is phosphorus (P) element, and a metalloid in the form of silicon (Si). In this study, some detected heavy metals, metals such as Cr and Co [12] were poisonous, even if they were present in small amounts. 

Aluminium (Al) is the abundant constituent in all collected soil samples. Its concentration in the samples range between 40,654 mg/kg and 21,943 mg/kg. High Al content was recorded on Site 1B soils, whereas Site 2 (denoted by D), a control site, carries low Al levels. The concentration of Al within the examined areas is as follows Site 1B > Site 1A > Site 1C > Site 1D. Al is not heavy in terms of atomic number and density but is emitted, transformed, and deposited through the atmosphere, similar to other heavy metals [13].

Calcium (Ca) also appears in all soil samples with the highest quantity values. The maximum Ca levels were found in Site 1A with a concentration value of 33,584 mg/kg, and the minimum Ca levels were at the control site (Site 2) with a small weight of 5286 mg/kg. Site 1A soils carry higher Ca concentrations than the other three sites, followed by Site 1C samples, and the lowest Ca amounts were noted on the control site (Site 2) soils.

Iron (Fe) is a major element present in this study and exists in large amounts in all samples. Its concentration ranges from 25,098 mg/kg to 15,339 mg/kg. Fe levels follow this descending order Site 1A > Site 1B > Site 2 > Site 1C.

Potassium (K) is present in large amounts for all examined samples. Its concentration is noted to range amongst 4773 mg/kg and 9717 mg/kg. The highest K concentration was found at Site 1B soils, followed by Site 1C, and the lowest values were recorded at Site 2 samples. Magnesium (Mg) existed at all sites in large quantities, and its maximum levels were found in Site 2 soils, whereas low amounts of Mg were noticed at Site 1C. The Mg concentration is recorded to range from 2055 mg/kg to 3704 mg/kg. 

Silicon (Si) is a predominating metalloid in all samples with a high concentration of 3933 mg/kg and the lowest values at 1698 mg/kg. Site 1B showed maximum Si values, while minimum amounts were noticed in Site 2 soils. The lowest concentration of elements was recorded below 100 mg/kg, those metals are Co with concentration values ranges between 9.0 mg/kg to17.0 mg/kg, Cr (35.0 mg/kg to 68.0 mg/kg), Ni (11.0 mg/kg to 21.0 mg/kg), Cu (17.0 mg/kg to 179.0 mg/kg), and Sr ranging from 33.0 mg/kg to 87.0 mg/kg. On the contrary, Gebeyehu and Bayisa 2020 noticed a high Co content, which was above the permissible limits in soil [14].

Mn, Na, Ba, and P have moderate concentrations, and their levels range between 169.0 mg/kg and 1029 mg/kg. Zinc (Zn) concentration is available in all soil samples and carries Zn levels ranging from 30.0 mg/kg recorded at the control site (Site 1D and 2D) and 1031 mg/kg noticed at Site 2A. The Zn levels randomly differ between the sampling sites.

**Table 2 ijerph-19-12054-t002:** Heavy metal concentration in soil analysed by ICP-MS.

Soil Heavy Metal Concentration (mg/kg)															
Sampling Sites	Al	Ba	Ca	Fe	K	Mg	Mn	Na	Ni	P	Sr	Zn	Co	Cr	Cu
Site 1A	21,943	265	8131	15,339	5835	1986	623	676	11	403	46	107	12	35	31
Site 1B	36,265	336	6348	22,608	9513	3139	656	523	15	420	50	160	11	45	82
Site 1C	25,198	228	8880	15,463	7162	2279	437	374	11	470	63	112	9	43	75
Site 1D	24,100	145	3149	17,267	5105	2125	417	496	18	296	25	30	12	57	12
Site 2A	29,435	402	33,584	25,098	9179	3704	938	1541	21	1344	87	1031	17	68	179
Site 2B	40,654	302	3431	23,227	9717	3006	527	363	15	284	36	65	11	47	24
Site 2C	24,435	213	5536	15,322	6989	2055	404	288	10	328	52	58	8	40	18
Site 2D	25,589	176	3651	18,707	4773	2419	489	587	19	192	33	30	13	63	12
Site 3A	25,893	356	7815	19,079	6913	2403	776	643	15	412	56	105	13	43	28
Site 3B	31,172	298	6335	19,629	8185	2670	633	476	15	346	47	129	11	44	33
Site 3C	26,855	232	7808	16,678	7066	2336	433	396	13	403	56	98	9	45	39
Site 3D	26,646	169	5286	20,163	5499	2759	499	1029	19	235	35	37	13	58	17
WHO permissible limits (mg/kg) [15]	n. a	n. a	n. a	1000	n. a	n. a	500	n. a	75	n. a	n. a	50	50	63	30

Note: n. a stand for not applicable.

### 3.2. Concentration of Selected Soil Enzymes Activity in the Soil

The variations in levels of enzymes activities are presented in Table 3 and graphically demonstrated in Figure 4. The invertase activity concentration ranges from 2.43 ± 1.52 to 3.66 ± 2.11 μg glucose⋅g^−^^1^ soil⋅h^−^^1^) and phosphatase (with a concentration range of 1.42 ± 0.82 and 3.98 ± 2.30 μg phenol⋅g^−^^1^ soil⋅h^−^^1^) were recorded in high amount for all soil samples, followed by the catalase activity content (ranging between 0.65 ± 0.37 and 2.63 ± 1.52 mL KMnO_4_⋅g^−^^1^ soil⋅h^−^^1^), and urease activity found in a minimal amount for all selected soil samples (with concentration range <1.00 μg NH_4_-N⋅g^−^^1^ soil⋅h^−^^1^. 

The obtained results show that the enzyme activity of invertase and phosphatase are suitable enzymes for soil remediation to reduce heavy metal pollution in the environment. 

### 3.3. The One-Way ANOVA Results of Soil Heavy Metals

The one-way ANOVA analysis of variance between heavy metals content showed a non-linear correlation (Table 4). There was a statistically significant difference between the determined heavy metals in the soil where F = 112.56 and *p* < 0.05 = 0.001. A Turkey post hoc analysis revealed that Al concentration is statistically significantly different from all the examined metals. In contrast, Barium (Ba) showed no statistically significant difference with *p* = 1.000 for selected metals (Co, Cr, Cu, K, Mg, Mn, Ni, Sr, and Zn). Although, Ca and Fe concentrations were statistically significantly different with Ba. 

Homogeneous subsets analysis shows that the mean levels of Co, Ni, Cu, Sr, Cr, Zn, Ba, Mn, Na, Mg are statistically the same. 

### 3.4. Correlation between Enzyme Activity and Metal Content in the Soil

The Pearson correlation analysis was conducted to determine the relationship between selected enzymes’ activity: urease, invertase, catalase, phosphatase, and heavy metals in soil (Table 5). The urease activity is associated negatively but non-significant with the concentration of Al, Ba, Ca, Co, Cu, Fe, K, Mg, Mn, Na, Sr, and Zn. While the amount levels of Ni and Cr indicated a positive non-significant association with urease activity. In comparison, a negative non-significant correlation existed between invertase and phosphatase enzyme activity with the concentration of all selected examined metals (Al, Ba, Ca, Co, Cu, Fe, K, Mg, Mn, Na, Ni, Cr, Sr, and Zn. 

The catalase enzyme activity showed a negative non-significant correlation with the concentration of Al, Co, Cr, Fe, K, Mg, Na, and Ni. However, the correlation was non-significantly positive amongst the catalase activity and amounts of Ba, Ca, Cu, Mn, Sr, and Zn.

## 4. Discussion

In this study, we managed to determine four different types of soil enzymes successfully. We noted the levels of urease activity and catalase activity in small amounts, while invertase and phosphatase enzymes we moderately high in all selected sample sites. Soil enzymes are natural molecules that catalyse soil microbial reactions [16] and primarily originate from microorganisms and some plants and animals’ residues [4]. Since enzyme activities play fundamental roles in soil chemical and biological reactions, enzymes accumulate as free enzymes, stabilized on clay surfaces and soil organic matter. The low levels of available enzyme activity in soil result from no plants and animals remaining in the site since the site is full of garbage containing different kinds of chemicals that can threaten the soil horizon. Based on this study’s finding, we noted that the two enzymes invertase and phosphatase, can monitor heavy metal pollution in soil. However, the correlation analysis proved that high rich metal content in soil could affect enzyme activity negatively by competing with the enzyme-substrate for the enzyme active site resulting in the enzyme becoming denatured and its activity hindered. The enzyme activities in society are affected negatively by high levels of heavy metals. Several studies published previously corroborate this study’s findings. Commonly, trace element toxicity correlates negatively with soil enzyme activity [4]. 

Catalase and phosphatase activities can be used as soil indicators of heavy metals contamination. The findings on soil enzyme activity show decreased catalase, alkaline, and acid phosphatase activities along the significant roadside [17]. These outcomes, as mentioned earlier, correspond to this study’s outcomes, where soil enzyme activity was in low amounts in Site 1 portions.

Soil enzyme inhibition depends on the concentration and the nature of heavy metals, and its levels vary from one enzyme to another. However, at a specific concentration, some heavy metals can enhance enzyme activity [18].

The urease activity decreases with arising heavy metal content due to chemical conformation changes because of coordination reactions. Based on Lewis’s hard, soft acids and base theory, the urease enzyme’s active sites are better coordinated when heavy metal contents are low, including thiol or imidazolyl groups [19].

Heavy metals indirectly affect soil enzymatic activities by shifting the microbial community, which synthesizes enzymes [20].

Angelovičová et al., 2014 noted that enzyme activities (urease, acid phosphatase, and alkaline phosphatase) significantly decreased with the increased heavy metal contents. They reported a significant negative correlation between urease activity and Zn content [21]. While Karaca et al., 2010 reported that the enzyme activities are influenced differently by various metals due to the different chemical affinities of the soil system’s enzymes. Pb significantly decreased urease, catalase, invertase, and acid phosphatase activities, while phosphatase and sulfatase were inhibited by arsenic (As) [16].

Another study by Al-Temimi et al., 2020 published a different trend, where the catalase enzyme activity is more effective in the soil’s K, Fe, Zn, and Mn concentrations. That is because K and other micronutrients are necessary to synthesize proteins and the enzyme’s effectiveness. K is a carrier element and acts to change the active site’s shape and configuration in the enzyme molecule better to bind the substrate and the enzyme [22].

Invertase and phosphatase enzyme activity seem suitable for monitoring soil pollution and soil management due to their high activity level recorded for this study. Soil enzymatic activity can diagnose the extent of soil function degradation caused by pollution and track the recovery of soil functions, even during the soil restoration process. Along with other biological and chemical features, enzyme activities may give important information on soil responses to poisoning and assist in determining if and to what degree the soil has regained its health [4].

The data acquired in this study highlights various environmental difficulties associated with landfills, including pollutant groundwater pollution, diffusion of wastes away from the site via surface run-off, aquifers, or emission into the atmosphere. 

As soon as soil is restored using the idea of sustainable remediation, the assessment of restoration efficiency may be appropriately carried out by monitoring changes in pollutants and metabolites and changes in soil functions before and after soil remediation [23]. As a result, it is critical to creating indicators that accurately track pollution decrease and soil quality recuperation. Soil enzymes, which are susceptible to management techniques and have been employed as markers of biogeochemical cycles, organic matter (OM) degradation, and soil remediation, can indicate ecological disturbances. When combined with other physical or chemical features, they can indicate soil quality. Because of their stability and sensitivity, soil enzymes can be utilized as biological indicators for evaluating soil quality; they can indicate if the biochemical reactions in the soil in which soil enzymes are engaged and done appropriately [4]. 

Proper monitoring of residents in the area can provide detailed information on the impact of such high heavy metal concentrations on their health. It can help draw future strategies to curb pollution. The necessary implementation of waste management programs will help the municipality adopt an approach that will promote recycling programs. The city needs to prioritize illegal dumping issues and provide physical facilities to manage waste.

## 5. Conclusions

Insufficient enzyme activity can result in an accumulation of chemicals that are harmful to the environment; some of these chemicals may further inhibit soil enzyme activity

Pollutant concentrations and soil enzyme activity have a negative connection in general. As a result, soil enzyme tests have frequently been used to differentiate contamination levels, giving important markers for identifying soil pollution. Up to this date, no single enzyme has been discovered as a universal indicator that can be utilized in various environmental situations, making it challenging to assess soil quality using soil enzymes [24].

Therefore, we can conclude that the soil from Alice landfill site is intimidating to many living organisms since it contains high heavy metal content, which has the potential of significantly causing damage to the physical and health conditions of humans and animal populations near the site. There is a need for biological agents and phytoremediation processes such as enzymes activity because they can transform and detoxify polluting substances effectively. They change pollutants at a detectable rate and potentially restore polluted environments. This can help decrease the high content of heavy metals, as the site is allocated not far from places where people stay.

## Figures and Tables

**Figure 1 ijerph-19-12054-f001:**
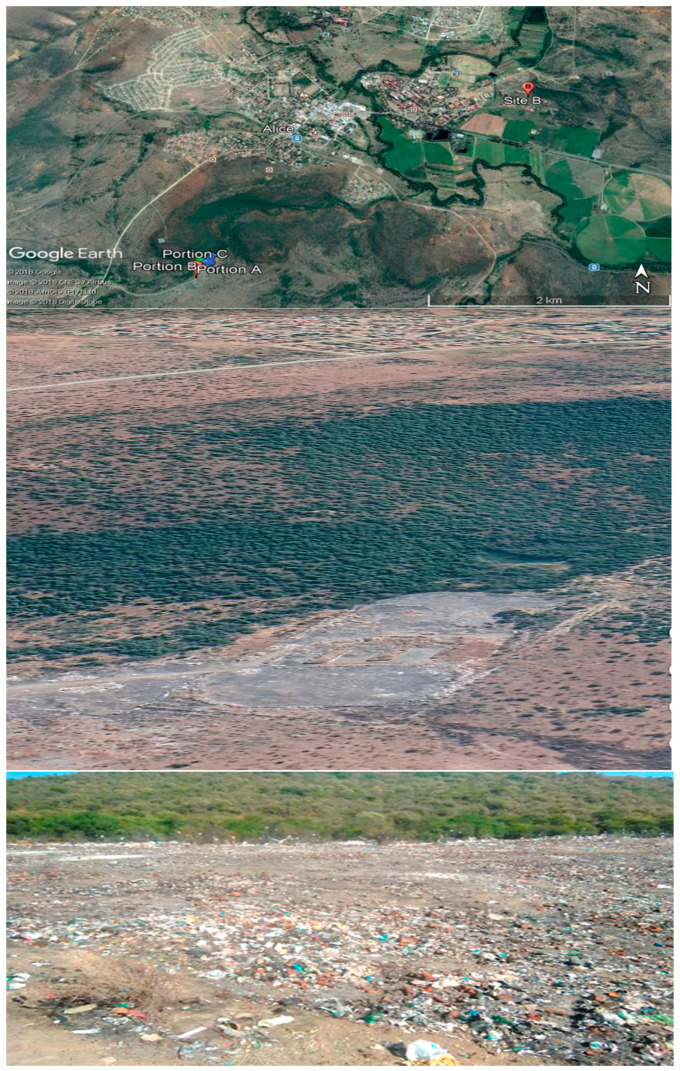
Pictures show the map of Alice, with the sampling sites and physical appearance of the landfill site.

**Figure 2 ijerph-19-12054-f002:**
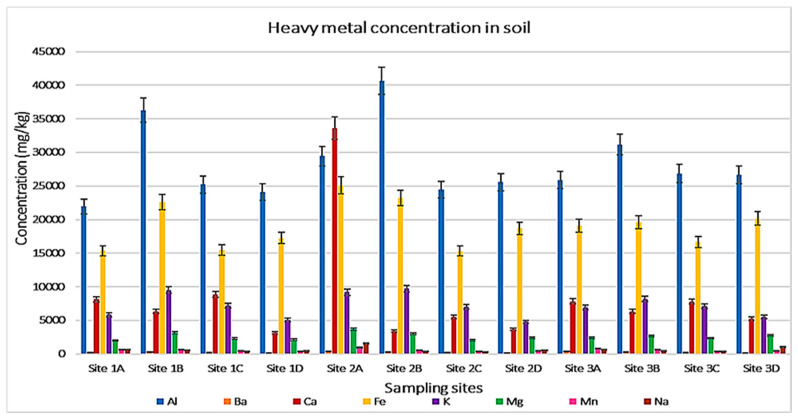
The results ICP-MS assessing the heavy metal concentration in soil samples.

**Figure 3 ijerph-19-12054-f003:**
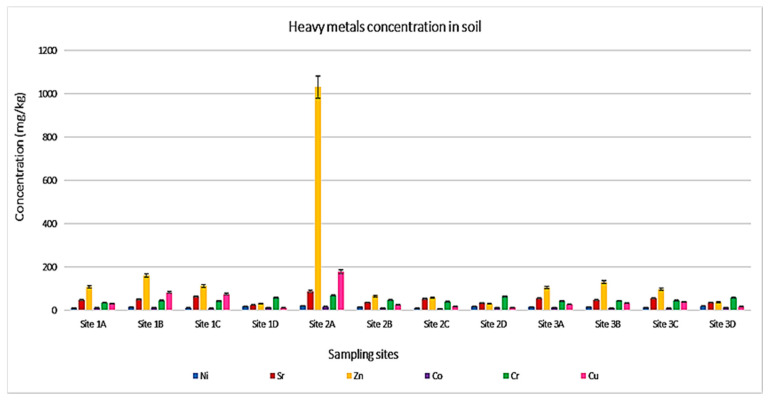
The results of ICP-MS assessing the heavy metal concentration in soil samples.

**Figure 4 ijerph-19-12054-f004:**
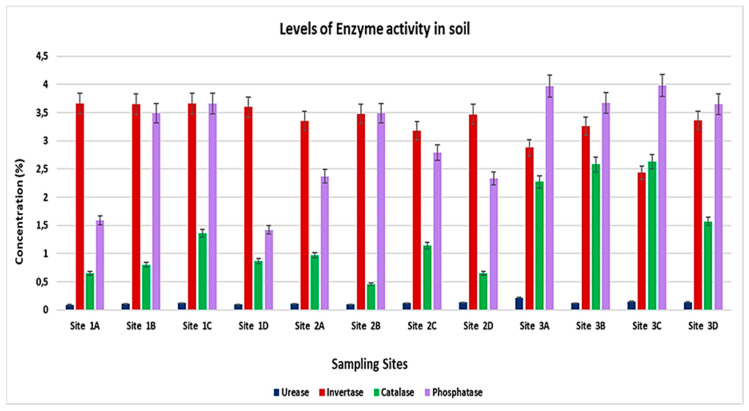
The levels of selected enzyme activity in different soil sampling sites.

**Table 1 ijerph-19-12054-t001:** Methods of soil enzyme activity assays.

Enzyme	Substrate	Incubation Hours	Metabolite
Invertase	3,5-Dinitrosalicylic acid	12	Glucose
Urease	Urea (CH_4_N_2_O)	3	NH_4_–N (Ammonium)
Catalase	3% H_2_O_2_ (Hydrogen peroxide)	3	KMnO_4_ (Potassium manganate)
Phosphatase	C_6_H_5_PO_4_Na_2_·2H_2_O (Phenyl phosphate disodium salt dihydrate)	1	Phenol (C_6_H_6_O)

**Table 3 ijerph-19-12054-t003:** The concentration of selected soil enzymes activity.

		Urease (μg NH_4_- N⋅g^−1^ Soil⋅h^−1^)	Invertase (μg Glucose⋅g^−1^ Soil⋅h^−1^)	Catalase (mL KmnO_4_⋅g^−1^ Soil⋅h^−1^)	Phosphatase (μg Phenol⋅g^−1^ Soil⋅h^−1^)
**Week 1**	Site 1A	0.09 ± 0.05 *^c^*	3.66 ± 2.11 *^d^*	0.65 ± 0.37 *^b^*	1.59 ± 0.92 *^d^*
	Site 1B	0.11 ± 0.06 *^c^*	3.65 ± 2.10 *^d^*	0.80 ± 0.46 *^cd^*	3.49 ± 2.01 *^ab^*
	Site 1C	0.12 ± 0.07 *^c^*	3.66 ± 2.11 *^d^*	1.36 ± 0.78 *^a^*	3.66 ± 2.11 *^e^*
	Site 2	0.10 ± 0.06 *^c^*	3.60 ± 2.08 *^d^*	0.87 ± 0.50 *^cd^*	1.42 ± 0.82 *^d^*
**Week 2**	Site 1A	0.11 ± 0.06 *^c^*	3.35 ± 1.93 *^d^*	0.97 ± 0.56 *^cd^*	2.37 ± 1.37 *^c^*
	Site 1B	0.10 ± 0.05 *^c^*	3.48 ± 2.01 *^d^*	0.46 ± 0.27 *^ab^*	3.49 ± 2.01 *^ab^*
	Site 1C	0.12 ± 0.07 *^c^*	3.18 ± 1.83 *^d^*	1.14 ± 0.66 *^a^*	2.79 ± 1.61 *^a^*
	Site 2	0.13 ± 0.07 *^c^*	3.47 ± 2.01 *^d^*	0.65 ± 0.37 *^b^*	2.33 ± 1.34 *^c^*
**Week 3**	Site 1A	0.22 ± 0.13 *^cd^*	2.88 ± 1.66 *^c^*	2.27 ± 1.31 *^c^*	3.97 ± 2.29 *^e^*
	Site 1B	0.12 ± 0.07 *^c^*	3.26 ± 1.88 *^d^*	2.58 ± 1.49 *^d^*	3.67 ± 2.12 *^e^*
	Site 1C	0.15 ± 0.09 *^c^*	2.43 ± 1.52 *^c^*	2.63 ± 1.52 *^d^*	3.98 ± 2.30 *^e^*
	Site 2	0.14 ± 0.0 *^c^*	3.36 ± 1.94 *^d^*	1.57 ± 0.91 *^a^*	3.65 ± 2.10 *^e^*

Results are presented as mean values ± SD; means with different letters within the same column show a significant difference (*p* < 0.05) at 95% interval. Letters a, b, c, and d in the means show that there is a statistically significant difference between each of the variables in the column.

**Table 4 ijerph-19-12054-t004:** The One-way ANOVA results of soil heavy metals.

ANOVA					
Metals					
	Sum of Squares	df	Mean Square	F	Sig.
Between Groups	11,565,655,644.36	13	889,665,818.797	112.556	0.000
Within Groups	1,217,252,533.27	154	7,904,237.229		
Total	12,782,908,177.63	167			

**Table 5 ijerph-19-12054-t005:** The correlation analysis between heavy metal concentration and enzyme activity in soil samples.

Correlations																		
Urease		Invertase	Catalase	Phosphatase	Al	Ba	Ca	Co	Cr	Cu	Fe	K	Mg	Mn	Na	Ni	Sr	Zn
**Urease**	1																	
**Invertase**	−0.013	1																
**Catalase**	0.202	−0.516	1															
**Phosphatase**	0.448	−0.230	0.754 **	1														
**Al**	−0.052	−0.067	−0.318	−0.315	1													
**Ba**	−0.343	−0.153	0.016	−0.282	0.480	1												
**Ca**	−0.265	−0.323	0.136	−0.218	−0.003	0.622 *	1											
**Co**	−0.139	−0.286	−0.140	−0.424	0.045	0.399	0.610 *	1										
**Cr**	0.044	−0.333	−0.369	−0.443	0.054	−0.075	0.448	0.707 *	1									
**Cu**	−0.117	−0.483	0.232	−0.071	0.205	0.675 *	0.921 **	0.478	0.369	1								
**Fe**	−0.136	−0.212	−0.379	−0.572	0.753 **	0.621 *	0.482	0.659 *	0.550	0.552	1							
**K**	−0.162	−0.108	−0.059	−0.216	0.818 **	0.786 **	0.390	0.018	−0.121	0.576 *	0.659 *	1						
**Mg**	−0.162	−0.251	−0.279	−0.475	0.684 *	0.654 *	0.639*	0.615 *	0.546	0.731 **	0.957 **	0.709 **	1					
**Mn**	−0.303	−0.218	0.006	−0.376	0.219	0.875 **	0.745 **	0.761 **	0.242	0.694 *	0.653 *	0.473	0.670 *	1				
**Na**	−0.208	−0.208	−0.033	−0.343	−0.068	0.385	0.806 **	0.893 **	0.664 *	0.665 *	0.571	0.073	0.642 *	0.717 **	1			
**Ni**	0.132	−0.243	−0.406	−0.485	0.171	0.062	0.387	0.831 **	0.931 **	0.324	0.682 *	−0.047	0.626 *	0.402	0.709 **	1		
**Sr**	−0.366	−0.202	0.175	−0.048	−0.012	0.702 *	0.854 **	0.230	0.038	0.853 **	0.245	0.506	0.453	0.631 *	0.481	−0.041	1	
**Zn**	−0.224	−0.351	0.081	−0.302	0.117	0.648 ^*^	0.986 **	0.645 *	0.510	0.928 **	0.592 *	0.460	0.724 **	0.765 **	0.804 **	0.466	0.796 **	1

Note: ** = Correlation is significant at the 0.01 level (2-tailed); * = Correlation is significant at the 0.05 level (2-tailed).

## Data Availability

The data presented in this study are available on request from the corresponding author.

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
