# Peer review of "Enzyme Activities in Reduction of Heavy Metal Pollution from Alice Landfill Site in Eastern Cape, South Africa"

_ijerph, 2022, doi:10.3390/ijerph191912054_

Round 1

Reviewer 1 Report

Comments to the Author:

This manuscript investigated the Enzyme Activities in reduction of heavy metal pollution. The objectives and experimental procedures are relatively clear, I suggest that it be revised before review. The article addressed the results of heavy metals in Eastern Cape South Africa. The topic of this paper is interesting, particularly because the heavy metals concentrations are changing in the soils due to anthropogenic activities. This modification in heavy metals inputs from anthropogenic sources could be affecting their availability for human and other organisms. However, I believe that the manuscript needs attention in some points.

1 The title:

It is not possible to identify what region of the world. I suggest to modify this to show more clearly where is the location. heavy metal pollution? I think the article referred to Ionization.

2 Introduction section:

Arrange your work in different paragraph according to the following order: 1) background of the study; 2) Statement of the problem (not list the biobibliography); 3) describe the gap of your research from previous studies; 4) Objectives of the study.

3 Material and methods:

  • A geological overview appeared to be essential for this type of study;
  • Justify the choice of these12 heavy metals?
  • the sampling map should be provided in the article.
  • surface soil 25cm?
  • The concentrations of Ni, Sr in the digested samples were determined using the ICP-MS. Generally, the concentrations of Ni, Sr in the fruits were too low to detected using ICP-MS. Provide the limit of detection.

4 Results and analysis

  • Figure 1 is not suitable.
  • Did the dataset follow the normal distribution. The dataset should follow the normal distribution before the correlation analysis.
  • does this factor correspond to the road network. I think the spatial distribution of normalized contribution for each factor should be provide to further demonstrate the results of ANOVA.
  • As well known, the outlier of the datasets may affect the results of ANOVA model. Many literatures eliminate the outliers to satisfied quality control. How does this article carry out data quality control?

5 Conclusions:

The conclusion should be improved.

Author Response

All the comments and corrections have been done 

Reviewer 2 Report

Evaluation and comments to manuscript ID ijerph-1442257, entitled “Enzyme Activities in reduction of heavy metal pollution from Alice dumpsite in Eastern Cape South Africa”

In my opinion, this is a very important manuscript from a public health point of view. Overall, the manuscript provides a new dataset, and the experimental design and data analysis are appropriate. However, I see a big problem in discussing the results. Despite this, the work may be a good contribution to the literature.

My comments:

1) the way of writing subscripts and superscripts in the formulas of chemical compounds,

2) description of table 3 (statistics) is at Fig. 3,

3) please use the correct method of citation in the text, e.g. ln 272 "... Aponte et al., 2020 reported ..."

4) Something is wrong, I get the impression that the end of the "Results" section is a discussion and the "discussion" section is a description of the results! Please establish it.

Author Response

The comments and corrections were properly done

Reviewer 3 Report

Comments:

The manuscript entitled “Enzyme Activities in reduction of heavy metal pollution from Alice dumpsite in Eastern Cape South Africa” is an interesting work. In addition, this article has some ambiguous statements which do not fully depict the information provided. Authors need to proofread the article thoroughly. The manuscript can be accepted for publication after major revisions. I suggest the following changes and improvements:

  1. The abstract is not well written. Authors need to revise it considering the key findings and remove the citation from the abstract section.
  2. Authors need to mention the full form of used abbreviation when they appear in the manuscript for the first time.
  3. Eliminate typos and grammatical issues throughout the manuscript.
  4. What is the research significance and novelty of this work?
  5. The soil samples were grinded ??
  6. All used chemicals should be mentioned with chemical names and formulas.
  7. Spaces need to be added between numbers and units (e.g., 0g, 5mL, 25˚C, 35.0mg/kg, etc.). By the way, thespace between the number and the percent sign can be omitted.
  8. The authors need to care about subscript (e.g., T HNO3, HClO4, etc.).
  9. Authors should revise the Conclusion Section considering the main findings. The current form of the Conclusion Section is not well written and attractive enough.

Author Response

All comments and corrections were attended and corrected

Round 2

Reviewer 2 Report

The authors responded to my comments, and the current version of the manuscript is significantly improved compared to the original version. Good job.